# A Qualitative PCR Assay for the Discrimination of Bubaline Herpesvirus 1, Bovine Herpesvirus 1 and Bovine Herpesvirus 5

**DOI:** 10.3390/microorganisms11030577

**Published:** 2023-02-24

**Authors:** Francesca Oberto, Emanuele Carella, Claudio Caruso, Pier Luigi Acutis, Davide Lelli, Luigi Bertolotti, Loretta Masoero, Simone Peletto

**Affiliations:** 1Istituto Zooprofilattico Sperimentale del Piemonte, Liguria e Valle d’Aosta (IZSPLV), Via Bologna 148, 10154 Torino, Italy; 2Azienda Sanitaria Locale CN1, Via Pier Carlo Boggio 12, 12100 Cuneo, Italy; 3Istituto Zooprofilattico Sperimentale della Lombardia e dell’Emilia Romagna, Via Bianchi 9, 25124 Brescia, Italy; 4Department of Veterinary Science, University of Turin, Largo Paolo Braccini 2, 10095 Grugliasco, Italy

**Keywords:** bubaline herpesvirus 1, bovine herpesvirus 1, bovine herpesvirus 5, infectious bovine rhinotracheitis, PCR

## Abstract

Bubaline herpesvirus 1 (BuHV-1), Bovine herpesvirus 1 (BoHV-1) and Bovine herpesvirus 5 (BoHV-5) are classified in the genus *Varicellovirus*, subfamily *Alphaherpesvirinae*. BoHV-1 is the causative agent of infectious bovine rhinotracheitis, BoHV-5 induces moderate disease in adult cattle while BuHV-1 has instead been associated with a decline in livestock production of water buffaloes. The aim of this study was to develop a qualitative PCR assay that allows the discrimination of BuHV-1, BoHV-1 and BoHV-5. The alignment of homologous genes identified specific nucleotide sequences of BuHV- 1, BoHV-1 and BoHV-5. The design of the primers and the optimization of the PCR assay were focused on the target sequences located on the portions of gD, gE and gG genes. This assay involved the use of three different PCR end-points: the PCR of a portion of the gD gene identified only the presence of BoHV-1; the PCR of a portion of the gE gene confirmed the presence of both BoHV-5 and BuHV-1; the PCR of a portion of the gG gene discriminated between BoHV-5 and BuHV-1, as the amplification product was observed only for BoHV-5. This qualitative PCR assay allowed the differentiation of BoHV-1 and BoHV-5 infections both in cattle and water buffaloes and heterologous BuHV-1 infections in bovine.

## 1. Introduction

The family Herpesviridae is divided into three subfamilies: *Alphaherpesvirinae*, *Betaherpesvirinae*, and *Gammaherpesvirinae*. The first subfamily contains five genera: *Simplexvirus*, *Varicellovirus*, *Mardivirus*, *Iltovirus* and *Scutavirus* [1]. The *Varicellovirus* genus of the family *Herpesviridae* includes nearly two hundred viruses isolated from various hosts [2,3,4,5,6]. The *Varicellovirus* genus includes a cluster of viruses antigenically and genetically related to bovine herpesvirus 1 (BoHV-1), such as bubaline herpesvirus 1 (BuHV-1), bovine herpesvirus 5 (BoHV-5), caprine herpesvirus 1 (CpHV-1), cervid herpesviruses 1 (CvHV-1) and 2 (CvHV-2), rangiferine herpesvirus 1 (RanHV-1) and elk herpesvirus 1 (ElkHV-1).

BoHV-1 is a double-stranded DNA virus and the etiological agent of the infectious bovine rhinotracheitis (IBR) that affects domestic and wild ruminants, [7]. Clinical signs involve both the respiratory system (viral infectious rhinotracheitis) and the reproductive system (pustular vulvovaginitis and infectious balanoposthitis), leading to spontaneous abortions and infertility. In addition, the virus is also known to cause immune suppression, conjunctivitis, encephalitis and a drop in milk production [8,9].

BoHV-5 is the primary etiological agent of meningoencephalitis in calves and has been shown to trigger fatal meningoencephalitis after intranasal inoculation or a meningitis-like disease after intravaginal infection [9,10,11]. Neurological signs start with severe depression and anorexia, jaw champing and hypersalivation.

Although BuHV-1 has been associated with subclinical disease in water buffalo (*Bubalus bubalis*), its pathogenic potential is still unclear. It was isolated for the first time from the prepuce of buffalo bulls in Australia. Later on, field strains of BuHV-1 were isolated from water buffaloes in Italy and Argentina [12,13]. In Italy, the water buffalo is important for the production of the traditional cheese “mozzarella di bufala” and is mainly raised in the center-south of the country, sometimes also with cattle [12,14].

The identification of BuHV-1, BoHV-1 and BoHV-5 in infected animals does not seem possible by serological methods, because their antigenic determinants are highly conserved among the ruminant alphaherpesviruses [15,16,17]. Consequently, the properties shared by those alphaherpesviruses can lead to a misdiagnosis of BoHV-1 infection, which is usually considered the main pathogen of infectious bovine rhinotracheitis [5,11,18]. An indirect ELISA kit, developed by Nogarol et al. in 2014, allows to discriminate BoHV-1 and BuHV-1 strains based on the expression of the glycoprotein E [19]. However, the test does not allow the differentiation between BoHV-5 and BuHV-1, due to the high similarity of the amino acid sequences of their glycoprotein E (97%) [12]. Moreover, BuHV-1 and BoHV-5 share 92.5% of the genome sequence and present the same nucleotide sequence of the gB gene [12,20,21]. Discrimination of BoHV-1 and BoHV-5 infections in cattle is also not possible by using commercial tests that detect total antibodies, due to a partial immunological cross-reactivity between these virus strains [20].

Biomolecular methods are usually employed for detecting ruminant alphaherpesviruses closely related to BoHV-1. The real-time PCR developed by Wang et al. in 2007, targeting the gB gene, detected most of the alphaherpesvirus strains, such as CvHV-1, CpHV-1, RanHV-1, BuHV-1, BoHV-1 and BoHV-5 [15], whereas other biomolecular methods have been developed to differentiate the different strains of alphaherpesvirus, except for BuHV-1 [16,22,23,24]. In 2016, Marin et al. designed a method to discriminate between BoHV-1 and BoHV-5 but was unable to discriminate BuHV-1 from other alphaherpesvirus strains [9]. In 2009, Campos et al. described a nested PCR method to determine the prevalence of BoHV-1 and BoHV-5 separately in the same samples [25]. Lastly, Ros et al. developed a specific diagnostic system in 1999 based on a nested PCR, which can be employed to detect and discriminate all the ruminant alphaherpesviruses closely related to BoHV-1, except for the BuHV-1 strain [24].

Under Council Directive 64/432/EEC (the directive on animal health problems affecting intra-community trade in bovine animals and swine), *Bison bison* and *Bubalis bubalis* are considered bovines. This has implications for the restriction of domestic animals in IBR control plans [26]. Cattle and water buffaloes that have tested positive for IBR, without differentiating among BuHV-1, BoHV-1 and BoHV-5 cannot be introduced in Italy. In fact, ruminant alphaherpesviruses are not always restricted to their natural host species. Indeed, water buffaloes, goat, sheep, red deer, roe deer, llamas, alpacas and camels were naturally or under experimental conditions infected with BoHV-1 [2,3,5,11,14,18,25,27,28,29]. This virus strain has also been reported in sheep with fatal pneumonia or abortion [22,27]. It was found in the trigeminal ganglia of experimentally infected water buffaloes, even if the virus reactivation was not observed. Conversely, cattle were shown to be susceptible to BuHV-1 infection even though the infected animals did not develop severe clinical signs [5,11,27,28,30]. Therefore, the increased interest in IBR eradication and control plans is related to a high probability of interspecies circulation of alphaherpesviruses in ruminants [31]. This leads to unjustified trade restrictions, reducing the economic income of the cattle industry. The aim of this study was the identification of discriminating genetic targets among BuHV-1, BoHV-1 and BoHV-5. This allowed the development of a qualitative PCR assay, that discriminate the different virus strains without using complex and time-consuming methods, such as genome sequencing.

## 2. Materials and Methods

### 2.1. Selection of Virus Strains

The BoHV-1 and BoHV-5 virus strains, with identification numbers 14/10/88 and Hb176/2 NA 67 Bartha, respectively, were kindly provided by the virology laboratory of Istituto Zooprofilattico Sperimentale della Lombardia e dell’Emilia Romagna. The BuHV-1 virus strain with identification number MR077 was kindly provided by Istituto Zooprofilattico Sperimentale del Mezzogiorno and has been isolated from water buffalo vaginal swab after dexamethasone treatment, as reported in a previous study [19].

### 2.2. Virus Propagation

The virus propagation was performed on permissive cell lines Madin Darby Bovine Kidney (MDBK) that were already available in our virology laboratory. A 25 cm^2^ flask of MDBK was prepared for each virus strain, containing Minimum Essential Medium with Earle’s Salts (MEM Earle medium) with 10% fetal bovine serum, 1% glutamate and 1% penicillin–streptomycin, to obtain a 90% confluent monolayer for the next day. Afterward the growth medium was removed from the flask, which was then infected with 1 mL of the virus strains suspended in fresh growth medium. The flask was incubated for 1 h at 37 °C in a thermostat equipped with a mechanical stirrer and then 7 mL of MEM Earle without fetal bovine serum was added. Lastly, the flask was incubated at 37 °C in a CO_2_ incubator.

Each infected flask was observed daily under the microscope to detect a possible cytopathic effect (CPE) induced by the virus characterized by the presence of rounded cells’ foci, with a tendency to the formation of syncytia and subsequent lysis of the cell monolayer. CPE induced by the BuHV-1, BoHV-1 and BoHV-5 was observed after 24 h post-infection. The flasks then underwent freeze–thaw cycles at least twice to allow the rupture of the cell membranes and the release of the virus into the extracellular environment. The suspensions of extracellular material and cells were centrifugated afterward at 3500 rpm for 20 min to collect the supernatant. The latter can be used for any subsequent passages or stored at −80 °C.

### 2.3. DNA Extraction

Before DNA extraction, the cell culture supernatant was treated with DNase I and purified with the DNA Clean & Concentrator-10 kit (Zymoresearch, Irvine, CA, USA). DNA was then extracted from 500 µL of cell culture supernatant using the DNeasy Blood & Tissue kit following the manufacturer’s protocol (Qiagen, Hilden, Germany).

### 2.4. Discrimination of BuHV-1, BoHV-1, and BoHV-5

An analysis of the reference sequences deposited in GenBank for BuHV-1 (GenBank accession no. NC043054.1), BoHV-1 (GenBank accession no. NC001847.1) and BoHV-5 (GenBank accession no. KY549446.1) was performed to detect possible specific genetic sequences. The alignment of homologous viral gene sequences, by using the CLUSTAL OMEGA program (https://www.ebi.ac.uk/Tools/msa/clustalo/; accessed on 14 July 2021), allowed the identification of specific nucleotide sequences of BuHV-1, BoHV-1 and BoHV-5. In silico analysis of the sequence alignments concerned different genes encoding the following proteins: BICP22, BICP0, BICP4, DNA Polymerase, gB, gC, gD, gE, gG, gH, gI, gL, gM, TK, UL6, UL5, UL8, UL9, UL12, UL13, UL14, UL16, UL17, UL18, UL19, UL20, UL22, UL23, UL25, UL26, UL27, UL28, UL29, UL48, UL52, US3, US9. The design of the primers and the optimization of PCR assay were focused on genetic targets located on the gD, gE and gG genes. The detection of amplified products for gD, gE and gG was performed by using end-point PCR on agarose gel (1.8%) [32].

#### 2.4.1. PCR of a Portion of gD Gene for Detecting BoHV-1

An end-point PCR has been performed to amplify the genomic target of the gD gene. The employed primers gD FW (5′-CCGCCGTATTTTGAGGAGTCG-3′) and gD RW (5′-TCGGTCTCCCCTTCRTCCTC-3′) were developed by Wernike et al. [33] and manufactured by Thermofisher. The PCR mixture consisted of 12.5 µL of Probe qPCR Mix, with UNG (Takara, Kusatsu, Japan), 200 nM of each primer, 5 µL of the template and 5.5 µL of DNase and RNase-free water, in a total volume of 25 µL. The PCR cycling conditions set on the Applied Biosystems 2720 Thermal Cycler (Life Technologies, Carlsbad, CA, USA) consisted of an initial step with a temperature of 50 °C for 2 min and a denaturation step with a temperature of 95 °C for 2 min, followed by 45 cycles at 95 °C for 15 s and at 60 °C for 45 s. Amplification products were analysed by the Amplisize Molecular Ruler (Biorad, Hercules, CA, USA).

#### 2.4.2. PCR of a Portion of gE Gene for Detecting BuHV-1 and BoHV-5

An end-point PCR was carried out to amplify the genomic target of the gE gene. The employed primers gE FW (5′-GAAGCCGACACCGGGGCAGAG-3′) and gE RW (5′-CGACGAACTCCCTGTCGCCCGCCGGA-3′) were manufactured by Thermofisher. The PCR mixture consisted of 12.5 µL of Probe qPCR Mix, with UNG (Takara), 400 nM of each primer, 5 µL of the template, 1 µL of dimethyl sulfoxide (DMSO) and 4.5 µL of DNase and RNase-free water, in a total volume of 25 µL. The PCR cycling conditions set on the Applied Biosystems 2720 Thermal Cycler (Life technologies) consisted of an initial step with a temperature of 50 °C for 2 min and a denaturation step with a temperature of 95 °C for 2 min, followed by 45 cycles at 95 °C for 15 s and at 61 °C for 45 s. Amplification products were loaded onto agarose gel (1.8%) and analysed by the Amplisize Molecular Ruler (Biorad).

#### 2.4.3. PCR of a Portion of gG Gene for Detecting BoHV-5

An end-point PCR was performed to amplify the genomic target of gG gene. The employed primers gG FW (5′-CTCGACCGGCGATTACG-3′) and gG RW (5′-GTGGCGTCACCACTACCAC-3′) were manufactured by Thermofisher. The PCR mixture consisted of 12.5 µL of Probe qPCR Mix, with UNG (Takara), 400 nM of each primer, 5 µL of the template and 5.5 µL of DNAse and RNase-free water, in a total volume of 25 µL. The PCR cycling conditions set on the Applied Biosystems 2720 Thermal Cycler (Life technologies) consisted of an initial step with a temperature of 50 °C for 2 min and a denaturation step with a temperature of 95 °C for 2 min, followed by 45 cycles at 95 °C for 15 s and at 60 °C for 45 s. Amplification products were analysed by the Amplisize Molecular Ruler (Biorad).

## 3. Results

The new qualitative PCR assay to discriminate BuHV-1, BoHV-1 and BoHV-5 was designed based on the results of in silico alignments of the portion of gD, gE and gG genes (Figure 1). Three end-point PCR were developed to discriminate the different virus strains. The PCR of a portion of the gD gene showed an amplification product of 110 bp only for BoHV-1 (Figure 2). The PCR of a portion of the gE gen resulted in an amplification product of 295 bp only for BuHV-1 and BoHV-5 (Figure 3). Finally, the PCR of a portion of the gG gene exhibited an amplification product of 296 bp only for BoHV-5 (Figure 4). Original and uncropped images of agarose gel electrophoresis are provided in the supplementary materials (Appendix A). In Table 1, the positivity of each gene to each virus and PCR product size is summarized.

## 4. Discussion

In the last decades, different biomolecular methods have been developed for the discrimination of ruminant alphaherpesviruses closely related to BoHV-1. Different multiplex PCR methods, both conventional and real-time methods, have been developed to discriminate between BoHV-1 and BoHV-5. Most of these diagnostic methods only discriminate between BoHV-1 and BoHV-5, while the remaining methods are capable to discriminate among nearly all BoHV-1-related alpha herpesviruses except BuHV-1. For example, Ros and Belak in 2001 reported a nested PCR combined with REA of the PCR products, to differentiate all the ruminant alphaherpesviruses, apart from BuHV-1 [29]. Instead, Keuser et al. in 2004 developed an immunofluorescence assay to discriminate the BoHV-1-related alphaherpesvirus, producing monoclonal antibodies (Mabs) specific for BoHV-1, BoHV-5, CpHV-1, CvHV-1 and CvHV-2. Unfortunately, this method was unable to detect viruses at latency sites in post-mortem samples but could only identify and discriminate the related ruminant alphaherpesviruses closely related to BoHV-1, after experimental reactivation and viral isolation [24]. Their protocols are very complex and time-consuming; therefore, they do not seem suitable for IBR routine testing.

In our study, the genetic targets with a greater degree of polymorphism, encoding the different proteins of BuHV-1, BoHV-1 and BoHV-5 were identified after carrying out in silico alignments. The selection focused on the genetic targets that presented suitable features for the development of a PCR protocol, such as the presence of polymorphisms among the virus strains, the design of primers with optimal length and the sequence and the optimal length of the amplification products. The specific genetic targets, useful for the development of an end-point PCR assay, are located on the portions of the gD, gE and gG genes. The amplification of the gene target gE was optimized by increasing the annealing temperature of the primers to 61 °C and adding 4% of DMSO to the reaction mix. These modifications are necessary to decrease the melting temperature of primers rich in GC regions and reduce the formation of non-specific bands, thus improving the amplification.

It was not possible to develop a multiplex PCR, either real-time or conventional, since some technical issues were encountered that led to the development of an end-point PCR assay for the discrimination of BoHV-1, BuHV-1 and BoHV-5. In fact, the high homology sequences have made it impossible to design specific probes within the identified gene targets, thus preventing the development of a real-time PCR assay [11,22,24]. Instead, the discriminating nucleotide sequences of BoHV-1, BuHV-1 and BoHV-5 present on the three viral genomes allowed to obtain amplification products of very similar size for the target sequences identified on gE and gG (respectively 295 bp and 296 bp), thus preventing their discrimination on agarose gel by using multiplex PCR.

According to the Italian IBR eradication and control plans, the trade of bovine and water buffalo that test positive for antibodies anti-IBR is not allowed, without discriminating BuHV-1, BoHV-1 and BoHV-5 [34]. BoHV-5 is mainly restricted to South America, especially Brazil and Argentina and is still absent in Italy, even though it could cross the national borders anytime [35]. Therefore, the presence of IBR in various ruminant species other than bovine species is a major threat to IBR eradication plans because confusion can arise when a bovine is falsely identified as BoHV-1 positive, even though it is infected with a related but distinct alphaherpesvirus. As this PCR assay clearly differentiates BuHV-1, BoHV-1 and BoHV-5, it could identify BoHV-1 and BoHV-5 infections both in cattle and water buffaloes and heterologous BuHV-1 infections in bovine. These findings could facilitate international trade by reducing the risk of introducing BuHV-1, BoHV-1 and BoHV-5 into countries where they have never been detected.

## Figures and Tables

**Figure 1 microorganisms-11-00577-f001:**
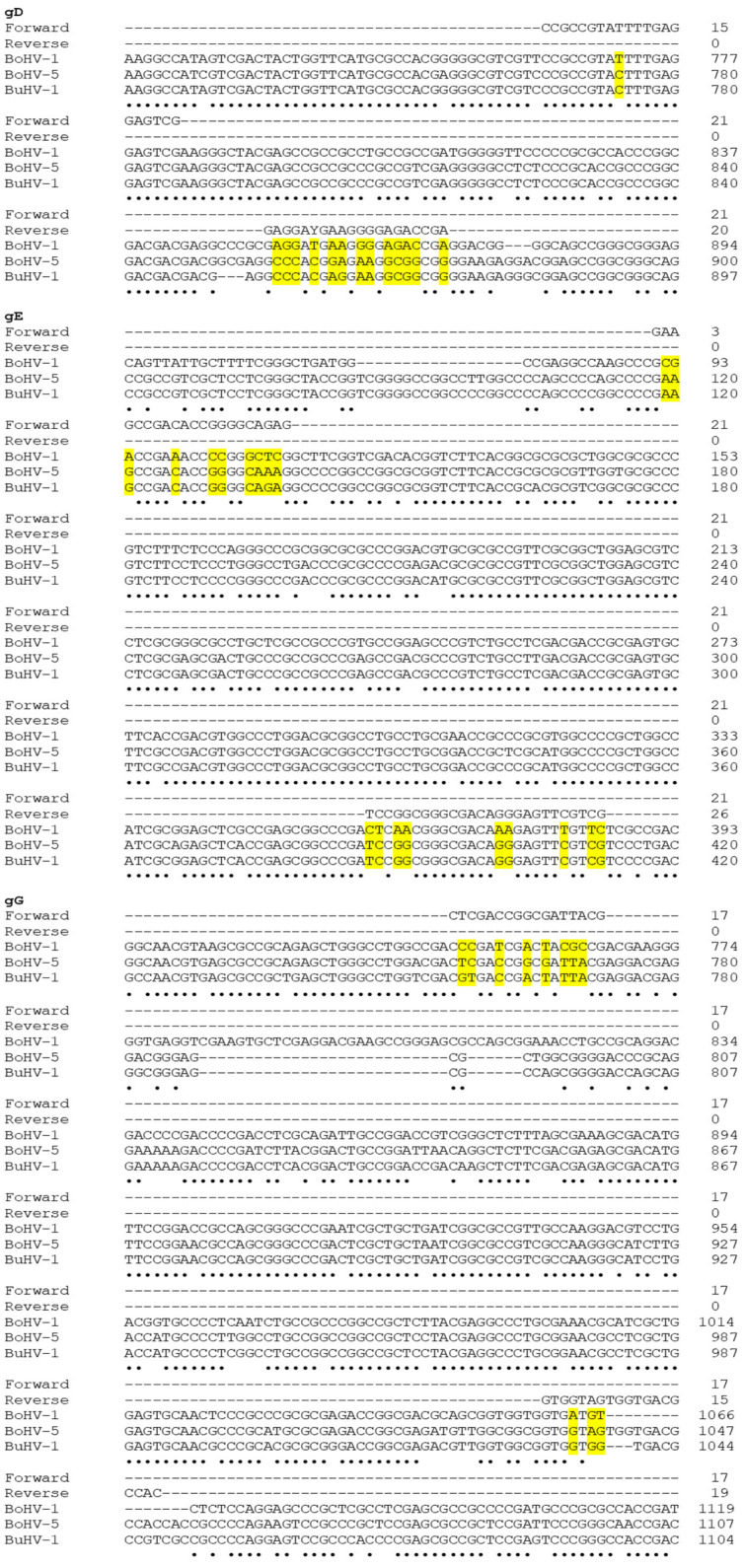
DNA sequence alignment for the portions of gD, gE and gG genes identified as targets to discriminate BuHV-1, BoHV-1 and BoHV-5. The symbol ‘•’ indicates monomorphic nucleotides. The yellow shadings indicate the polymorphic nucleotides (SNPs) at the primer attachment sites. The number of bases depicted in each line is marked by the number shown at the right.

**Figure 2 microorganisms-11-00577-f002:**
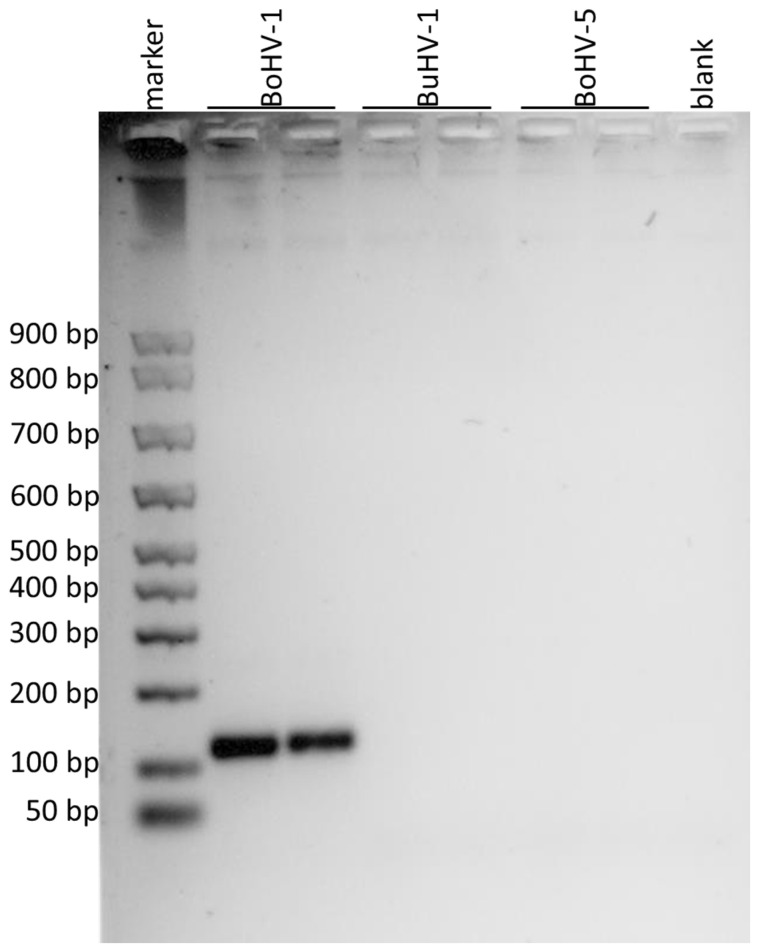
Agarose gel electrophoresis of the amplified products obtained by using the primers for the portion of gD gene. The DNA marker in the first lane showed the first band at 50 bp, the second at 100 bp and the subsequent ones at 100 bp from each other. BoHV-1 was loaded in the second and third lanes, BuHV-1 in the fourth and fifth, BoHV-5 in the sixth and in the seventh and the eighth lane was the negative control (blank). Amplification products of 110 bp were observed for BoHV-1.

**Figure 3 microorganisms-11-00577-f003:**
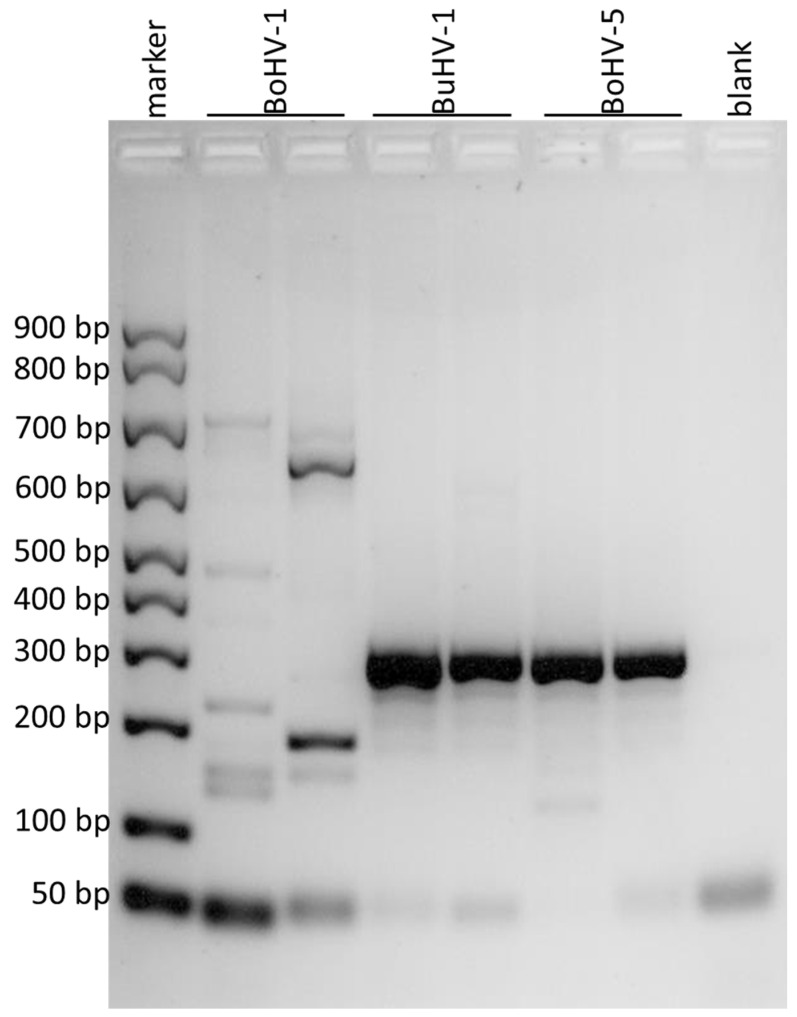
Agarose gel electrophoresis of the amplified products obtained by using the primers for the portion of gE gene. The DNA marker in the first lane showed the first band at 50 bp, the second at 100 bp and the subsequent ones at 100 bp from each other. BoHV-1 was loaded in the second and third lanes, BuHV-1 in the fourth and fifth, BoHV-5 in the sixth and in the seventh and the eighth lane was the negative control (blank). Amplification products of 295 bp were observed for BuHV-1 and BoHV-5.

**Figure 4 microorganisms-11-00577-f004:**
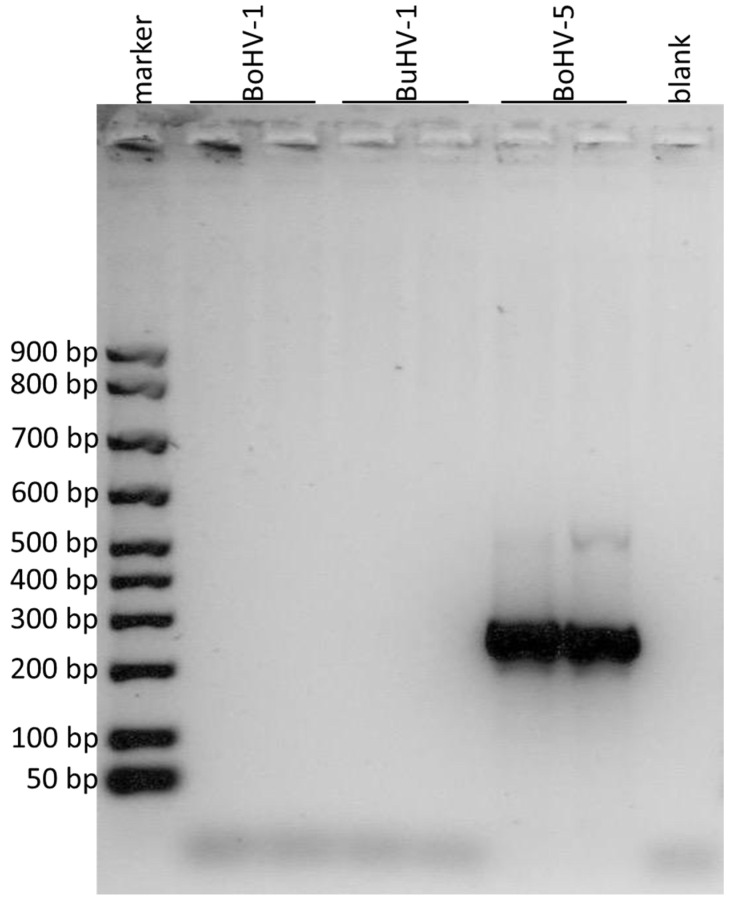
Agarose gel electrophoresis of the amplified products obtained by using the primers for the portion of gG gene. The DNA marker in the first lane showed the first band at 50 bp, the second at 100 bp and the subsequent ones at 100 bp from each other. BoHV-1 was loaded in the second and third lanes, BuHV-1 in the fourth and fifth, BoHV-5 in the sixth and in the seventh and the eighth lane was the negative control (blank). Amplification products of 296 bp were observed for BoHV-5.

**Table 1 microorganisms-11-00577-t001:** Summary table of the results obtained by using the PCR assay.

Virus Strains	gD Gene	gE Gene	gG Gene
BuHV1		295 bp	
BoHV1	110 bp		
BoHV5		295 bp	296 bp

## Data Availability

The data presented in this study are available on request from the corresponding author.

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
