# Peer review of "A Qualitative PCR Assay for the Discrimination of Bubaline Herpesvirus 1, Bovine Herpesvirus 1 and Bovine Herpesvirus 5"

_microorganisms, 2023, doi:10.3390/microorganisms11030577_

Round 1

Reviewer 1 Report

The manuscript titled "Development of a diagnostic method for the identification and discrimination of Bubaline herpesvirus 1, Bovine herpesvirus 1 and Bovine herpesvirus 5" describes a diagnostic tool to differentiate samples respecting the presence of BuHV-1, BoHV-1, and BoHV-5. The objective of the work is meaningful considering that three of those herpesviruses might be present in bovine or buffalo samples. However, the authors don't effectively describe the diagnostic route to classify the samples. Thus, there are some major and some minor concerns to take into account, and English should be revised by a native.

Major concerns:

1-   As it was mentioned before, the authors don’t clearly describe the diagnostic flow to consider a sample positive for any of the viruses. They mention neither in the abstract nor in the introduction that the diagnostic route is three end-point PCRs. When they mention a rapid diagnostic system other techniques than end-point PCRs come to mind.

2-   The materials and methods section is confusing. Why do the authors describe library preparations? To sequence the virus? That’s not well explained and table 2 is not self-explained. Why did the authors want to sequence the whole virus? They state in silico analysis of many genes but they describe the use of gB, gD, gE, and gG only. The authors don’t explain why they finally use those genes.

3-   The results section is not well described. The aim of the work is still not understandable and the virus classification criteria are not well explained. The use of gB PCR is not well explained, is the qPCR included in the diagnostic route? and gD, gE, and gG PCRs are not connected with the virus classification. I suggest adding a table in which the authors point the positivity of each gene to each virus and PCR product size to better understand which virus is positive to which gene PCR and quickly visualize which virus is in the tested sample.

4-   gG is missing and gE reverse primer is missing in figure 2.

5-   The authors work with highly purified viral batches. It is highly recommended to work with animal samples to validate a diagnostic tool. Compare it with other available techniques and demonstrate that other viruses are not detected with that assay. I am aware that this is not a statistically validated diagnostic tool but if the authors titled Development of a diagnostic method they must run some real samples at least. Which is the detection limit? Are the assays sensitive enough when using biological samples?

6-   The discussion section describes many diagnostic techniques that had been developed for BoVH-1 and 5 but not together with BuHV-1. Why is this a major concern? Some were mentioned in the introduction but they should be discussed.

Minor concerns:

1-   Abstract: the name of the technique should be stated. Results should be better explained.

2-   Line 35: citation 1 is too new. Please add the ICTV taxonomy link.

3-   Line 114: positive to antigen or antibodies?

4-   Line 119: how fast is the entire diagnostic pathway? The authors should tell how long it takes.  

5-   Line 165: to determine viral DNA concentration authors performed a qPCR for all viruses (again not well explained). How much was that concentration? Table 1 depicted Ct values but not concentrations.

6-   Section 2.5 should be 2.4.1; section 2.6 should be 2.4.2 and second section  2.6 should be 2.4.3

7-   Line 221: fraction of gE instead of gD

8-   Line 234: and gG instead of gD

9-   When gD, gE, and gG PCRs are described, viral strain specificity should be mentioned. As gD PCR was designed for detecting BoHV-1; gE PCR was designed for detecting BuHV-1 and BoHV-5, and gG PCR was designed for the detection of BoHV-5

10-               Line 248: Ct value

11-               Figure 1 and line 254: MDCK cells?? M&M describes MDBK cells

12-               Figure 1: 10X? a 10x objective plus a 10x eyepiece = 100x magnification.

13-               Line 275: Is gB included in the diagnostic route?

14-               Figure 2 is confusing. It would be better to show the similarities as asterisks or points so differences between genes are more visible. The end of gE and gG genes are missing

15-               Figure 3 should be re-designed. The molecular weight of the marker should be added to the figure as well as the name of the samples. What is in the eighth lane? Please describe or cut it. 110 bp product should be described in the figure caption.

16-               Figure 4: The molecular weight of the marker should be added to the figure as well as the name of the samples. What is in the eighth lane? Please describe or cut it.

17-               Figure 5: The molecular weight of the marker should be added to the figure as well as the name of the samples. What is in the eighth lane? Please describe or cut it.

18-               Line 341: another herpesvirus can be detected with the qPCR mentioned.

Author Response

1-   As it was mentioned before, the authors don’t clearly describe the diagnostic flow to consider a sample positive for any of the viruses. They mention neither in the abstract nor in the introduction that the diagnostic route is three end-point PCRs. When they mention a rapid diagnostic system other techniques than end-point PCRs come to mind.

The abstract and the introduction have been corrected according to your comments

2-   The materials and methods section is confusing. Why do the authors describe library preparations? To sequence the virus? That’s not well explained and table 2 is not self-explained. Why did the authors want to sequence the whole virus? They state in silico analysis of many genes but they describe the use of gB, gD, gE, and gG only. The authors don’t explain why they finally use those genes.

The section of materials and methods regarding the sequencing of virus strains has been deleted together with Figure 1 and table 1 and 2, as it is unnecessary for the purposes of this research. The design of the primers and the optimization of PCR assay were focused on genetic targets located on the gD, gE, gG genes. They present nucleotide sequences with a greater degree of polymorphism capable of differentiating the three virus strains

3-   The results section is not well described. The aim of the work is still not understandable and the virus classification criteria are not well explained. The use of gB PCR is not well explained, is the qPCR included in the diagnostic route? and gD, gE, and gG PCRs are not connected with the virus classification. I suggest adding a table in which the authors point the positivity of each gene to each virus and PCR product size to better understand which virus is positive to which gene PCR and quickly visualize which virus is in the tested sample.

The Real-Time PCR assay, that amplifies the target on gB gene, has been deleted from the manuscript, as it is unnecessary for the purposes of this research. A table to better understand which virus is positive to which gene has been added to the manuscript

4-   gG is missing and gE reverse primer is missing in figure 2.

An incomplete figure 2 was uploaded by mistake. Figure 2 with gG and gE reverse primer has been uploaded

5-   The authors work with highly purified viral batches. It is highly recommended to work with animal samples to validate a diagnostic tool. Compare it with other available techniques and demonstrate that other viruses are not detected with that assay. I am aware that this is not a statistically validated diagnostic tool but if the authors titled Development of a diagnostic method they must run some real samples at least. Which is the detection limit? Are the assays sensitive enough when using biological samples?

The aim of this study was to identify DNA sequences of BoHV-1 BuHV-1 and BoHV-5 to be used for their discrimination by using a PCR assay. For this reason, neither the limit of detection nor the sensitivity has been determined. Therefore, the title and the type of the manuscript have been changed, as only qualitative analysis was carried out.

6-   The discussion section describes many diagnostic techniques that had been developed for BoVH-1 and 5 but not together with BuHV-1. Why is this a major concern? Some were mentioned in the introduction but they should be discussed.

The discussion has been improved in the manuscript, according to your comment

Minor concerns:

1-   Abstract: the name of the technique should be stated. Results should be better explained.

The Abstract has been modified according to your comment

2-   Line 35: citation 1 is too new. Please add the ICTV taxonomy link.

The citation 1 has been deleted and the ICTV taxonomy link has been added

3-   Line 114: positive to antigen or antibodies?

Antibodies. The sentence has been modified and move in the discussion section

4-   Line 119: how fast is the entire diagnostic pathway? The authors should tell how long it takes. 

The adjective "rapid" to describe the PCR assay has been deleted from the manuscript.

5-   Line 165: to determine viral DNA concentration authors performed a qPCR for all viruses (again not well explained). How much was that concentration? Table 1 depicted Ct values but not concentrations.

Figure 1, Table 1, Table 2, and the description of next-generation sequencing have been deleted, since they are unnecessary for the purposes of this research

6-   Section 2.5 should be 2.4.1; section 2.6 should be 2.4.2 and second section  2.6 should be 2.4.3

The sections have been modified in the manuscript

7-   Line 221: fraction of gE instead of gD

The mistake has been corrected in the manuscript

8-   Line 234: and gG instead of gD

The mistake has been corrected in the manuscript

9-   When gD, gE, and gG PCRs are described, viral strain specificity should be mentioned. As gD PCR was designed for detecting BoHV-1; gE PCR was designed for detecting BuHV-1 and BoHV-5, and gG PCR was designed for the detection of BoHV-5

Virus strain specificity has been mentioned in the material and method section 2.3.1, 2.3.2, and 2.3.3

10-               Line 248: Ct value

The table 1 has been deleted from the manuscript, since it is unnecessary for the purposes of this research

11-               Figure 1 and line 254: MDCK cells?? M&M describes MDBK cells

The sentence in line 254 has been inserted in material and method section. Figure 1 has been deleted from the manuscript, since it is unnecessary for the purposes of this research

12-               Figure 1: 10X? a 10x objective plus a 10x eyepiece = 100x magnification.

The figure 1 has been deleted from the manuscript, since it is unnecessary for the purposes of this research

13-               Line 275: Is gB included in the diagnostic route?

gB was not included in the diagnostic route. The Real-Time PCR assay, that amplifies the target on gB gene, has been deleted from the manuscript, as it is unnecessary for the purposes of this research. The sentence in line 275 has been corrected

14-               Figure 2 is confusing. It would be better to show the similarities as asterisks or points so differences between genes are more visible. The end of gE and gG genes are missing

The Figure 2 has been re-designed according to your comment

15-               Figure 3 should be re-designed. The molecular weight of the marker should be added to the figure as well as the name of the samples. What is in the eighth lane? Please describe or cut it. 110 bp product should be described in the figure caption.

The Figure 3 has been re-designed according to your comment

16-               Figure 4: The molecular weight of the marker should be added to the figure as well as the name of the samples. What is in the eighth lane? Please describe or cut it.

The Figure 4 has been re-designed according to your comment

17-               Figure 5: The molecular weight of the marker should be added to the figure as well as the name of the samples. What is in the eighth lane? Please describe or cut it.

The Figure 5 has been re-designed according to your comment

18-               Line 341: another herpesvirus can be detected with the qPCR mentioned.

The sentence has been corrected and moved in the introduction section.

Reviewer 2 Report

Re: Microorganisms Manuscript ID: microorganisms-2160370

This manuscript described a new diagnostic method for identifying and discriminating bubaline herpesvirus 1, bovine herpesvirus 1, and bovine herpesvirus 5. The method is helpful for those purposes. However, there are many concerns to be resolved as follows. 

General comments.

1. The manuscript's results need to include more sufficient data as an Article. It would help if you changed the manuscript to 'Short Communication.'

2. The Introduction section is very verbose. Also, the Discussion section includes contents suitable to the Introduction section. Please describe it as compact as possible.

3. In the Results section,

Figure 1, Table 1, and Table 2 should be omitted. Readers know the CPE of BoHV-1. What is the purpose of the description of Ct values in Table 1? In this manuscript, the Ct value is not needed. What is the purpose of the description of next-generation sequencing? The data is unnecessary in the manuscript.

4. In Figure 2,

The data on the gG gene is missing.

Minor Points

Lines 43 and others: We do not use BHV-1.3 (maybe, BoHV-1.3). BoHV-1 is divided into three subtypes, BoHV-1.1, BoHV-1.2a, and BoHV-1.2b.

Lines 52-53 and 54-56: The descriptions concerning BoHV-1.3 and BoHV-5 would be the same.

Lines 147-148: What is the meaning of ‘freeze-thaw cycles for at least 24 h’?

Line 171: Please describe the name of the instrument for RT-PCR.

Line 221: ‘gD’ would be ‘gE.’

Line 234: ‘gD would be ‘gG.’

Lines 208-242: The three methods should be described in one method section.

Line 287: Why the intensity of the bands of lanes 2 and 3 are entirely different?

Line 300: BuHV-1 would be incorrect. ‘the viruses’ or ‘BoHV-1 or BoHV-5’ might be suitable instead of BuHV-1. 

Author Response

General comments.

  1. The manuscript's results need to include more sufficient data as an Article. It would help if you changed the manuscript to 'Short Communication.'

The manuscript has been changed as Short Communication

  1. The Introduction section is very verbose. Also, the Discussion section includes contents suitable to the Introduction section. Please describe it as compact as possible.

The discussion and the introduction have been modified according to your comment

  1. In the Results section,

Figure 1, Table 1, and Table 2 should be omitted. Readers know the CPE of BoHV-1. What is the purpose of the description of Ct values in Table 1? In this manuscript, the Ct value is not needed. What is the purpose of the description of next-generation sequencing? The data is unnecessary in the manuscript.

Figure 1, Table 1, Table 2, and the description of next-generation sequencing have been deleted since they are unnecessary for the purposes of this research.

  1. In Figure 2, The data on the gG gene is missing

An incomplete figure 2 was uploaded by mistake. Figure 2 with gG and gE reverse primer has been uploaded in the manuscript

Minor Points

Lines 43 and others: We do not use BHV-1.3 (maybe, BoHV-1.3). BoHV-1 is divided into three subtypes, BoHV-1.1, BoHV-1.2a, and BoHV-1.2b.

The introduction has been corrected according to your comments

Lines 52-53 and 54-56: The descriptions concerning BoHV-1.3 and BoHV-5 would be the same.

The description concerning the subtypes of BoHV-1.3 has been deleted

Lines 147-148: What is the meaning of ‘freeze-thaw cycles for at least 24 h’?

The sentence has been corrected in the manuscript

Line 171: Please describe the name of the instrument for RT-PCR.

The description of next-generation sequencing has been deleted from the material and methods

Line 221: ‘gD’ would be ‘gE.’

The mistake has been corrected in the manuscript

Line 234: ‘gD would be ‘gG.’

The mistake has been corrected in the manuscript

Lines 208-242: The three methods should be described in one method section.

The section in the material and method has been modify according to your comment

Line 287: Why the intensity of the bands of lanes 2 and 3 are entirely different?

The intensity of the bands of lanes 2 and 3 are entirely different, probably due to a degradation of DNA template in the lane 3 leading to a decreased intensity of the band. However, the PCR on gD gene portion has been repeated and the new figure has been uploaded to the manuscript.

Line 300: BuHV-1 would be incorrect. ‘the viruses’ or ‘BoHV-1 or BoHV-5’ might be suitable instead of BuHV-1.

The method reported by Marin et al. was unable to discriminate BuHV-1 from other alphaherpesvirus strains, such as BoHV-1 and BoHV-5. However, the sentence has been corrected in the manuscript.

Round 2

Reviewer 2 Report

Please consider the following points.

Major Points

1. Your modification of the Introduction and Discussion needs to be revised to understand your work. The discussion section still contains many descriptions suitable in Introduction. For example, lines 208 to 253 were contents to describe in the Introduction and describe them as concisely as possible. Reconstruction of the Introduction and Discussion is necessary to improve your manuscript.

2. The Object of your PCR assay development needed to be adequately described in the Introduction. Please explain it clearly.  

3. Preparation of template DNAs for the PCR assay must be included in Materials and Methods.

Minor Points

1. line 107: ‘Virus isolation’ would be ‘Virus propagation.’

2. line 110: ‘Minimun’ would be ‘Minimum.’

3. lines 143 and 144: Why did you choose R nucleotides instead of A in the 15th position of the gD RW primer sequence?

4. line 245: ‘murine antibodies’ would be ‘monoclonal antibodies.

5. Figure 1

A description of the numbers on the right hand is needed.

6. Figure 3

The nonspecific bands were observed in both BoHV-1 lanes. These results were inconsistent with your mentions on lines 260 to 265. The PCR condition would be reconsidered.
